# Secondary Metabolites of The Endophytic Fungus *Alternaria alternata* JS0515 Isolated from *Vitex rotundifolia* and Their Effects on Pyruvate Dehydrogenase Activity

**DOI:** 10.3390/molecules24244450

**Published:** 2019-12-04

**Authors:** Changyeol Lee, Wei Li, Sunghee Bang, Sun Joo Lee, Nam-young Kang, Soonok Kim, Tae In Kim, Younghoon Go, Sang Hee Shim

**Affiliations:** 1College of Pharmacy, Duksung Women’s University, 144 Gil 33, Dobong-gu, Seoul 01369, Korea; jaber29@naver.com (C.L.); scbsh4331@hanmail.net (S.B.); 2Korean medicine (KM)-Application Center, Korea Institute of Oriental Medicine (KIOM), Daegu 41062, Korea; liwei1986@kiom.re.kr (W.L.); tikim@kiom.re.kr (T.I.K.); 3New Drug Development Center, Daegu-Gyeongbuk Medical Innovation Foundation, 80 Cheombok-ro, Dong-gu, Daegu 41061, Korea; disjrk@dgmif.re.kr; 4Department of Creative IT Engineering, Pohang University of Science and Technology (POSTECH), 77 Cheongam-ro, Namgu, C5 building, room203, Pohang, Kyungbuk 37673, Korea; knysg@postech.ac.kr; 5Biological Resources Assessment Division, National Institute of Biological Resources, Incheon 22689, Korea; sokim90@korea.kr

**Keywords:** *Alternaria alternata*, endophytes, pyruvate dehydrogenase

## Abstract

The fungal strain *Alternaria alternata* JS0515 was isolated from *Vitex rotundifolia* (beach vitex). Twelve secondary metabolites, including one new altenusin derivative (**1**), were isolated. The isolated metabolites included seven known altenusin derivatives (**2**–**8**), two isochromanones (**9**, **10**), one perylenequinone (**11**), and one benzocycloalkanone (**12**). Their structures were determined via 1D and 2D nuclear magnetic resonance (NMR) spectroscopy, mass spectrometry (MS), and computational electronic circular dichroism (ECD) analysis. Compounds **3** and **11** increased pyruvate dehydrogenase (PDH) activity in AD-293 human embryonic kidney cells and significantly inhibited PDH phosphorylation. The IC_50_ values of **3** and **11** were 32.58 and 27.82 μM, respectively.

## 1. Introduction 

Endophytes are microorganisms that live within the internal tissues of plants, and they form symbiotic relationships with their host plants [1]. The functional diversity of endophytic fungi is notable among plant-associated microbes [2,3]. Endophytic fungi have been identified as sources of various bioactive metabolites with interesting structures, which are potential candidates for drug development [4,5,6]. Endophytic fungi have been reported to protect their host plants by producing diverse biologically active secondary metabolites with antiviral, antifungal, and antibacterial properties [7]. 

Halophytes are plants that have adapted to growing in highly saline water, and they comprise only 2% of all plant species [8]. The relationships between halophytes and their endophytes could help the plants adapt to highly saline conditions [9]. *Vitex rotundifolia* Linne fil. (Verbenaceae) is a halophyte that is widely distributed along the coast of East Asia [10]. Various chemicals have been isolated from *V. rotundifolia*, including lignans, diterpenes, lactones, glycerols, flavonoids, and iridoids [11]. Its fruit has been used as a folk remedy to treat asthma, chronic bronchitis, colds, ocular pain, female hormonal imbalance, headaches, migraines, and gastrointestinal infections [12,13]. Previous studies have identified a range of bioactivities of *V. rotundifolia*, including antioxidative, anticancer, and antiproliferative activity [14,15].

The endophytic fungi *Cochliobolus geniculatus*, *Curvularia* sp., *Nemania primolutea*, *Paecilomyces* sp., *Phoma* sp., and *Nemania primolutea* have been isolated from the leaves of *V. rotundifolia* grown in the coastal regions of the Malaysian Peninsula. In particular, *C. geniculatus, Curvularia* sp., *Paecilomyces* sp., and *Phoma* sp. exhibit antibacterial activity [16]. An endophytic fungus isolated from rhizomes of *V. rotundifolia* grown in the coastal region of Korea has a growth-promoting effect in Waito-C rice [17]. 

*Alternaria alternata* JS0515 was among the first reported endophytic fungi isolated from *V. rotundifolia* rhizomes. *A. alternata* JS0515 is found widely in nature [18]. Previous chemical investigations of *A. alternata* JS0515 identified phenolics, pyranones, quinones, steroids, terpenoids, and nitrogen-containing metabolites, some of which exhibited phytotoxic, cytotoxic, antifungal, and antimicrobial activities [19,20,21,22].

The pyruvate dehydrogenase complex (PDC) is a multienzyme complex and a crucial metabolic gatekeeper: it is the convergence point between glycolysis and the tricarboxylic acid (TCA) cycle for ATP generation. Its pyruvate dehydrogenase (PDH) E1α subunit catalyzes the oxidative decarboxylation of pyruvate into acetyl-CoA in the mitochondria [23,24,25,26,27]. PDH E1α activity is inhibited by the phosphorylation of its serine residues. Suppression of PDH activity is associated with various metabolic disorders, including obesity, non-alcoholic fatty liver disease, diabetes, and cancer [28,29,30,31,32,33]. In this study, twelve secondary metabolites were isolated from the ethyl acetate extracts of *A. alternata* (JS0515). The twelve secondary metabolites were then evaluated as PDH activators in a cellular PDH activity assay using AD-293 cells. We are thankful to the reviewers for reminding us of this crucial information. The human AD-293 cell line is a derivative of the commonly used HEK293 cell line. HEK293 cell line is often used in the inhibition of pyruvate dehydrogenase.

## 2. Results and Discussion

### 2.1. Isolation and Structural Elucidation

Twelve secondary metabolites, including eight altenusin derivatives (**1**–**8**), two isochromanones (**9**, **10**), one perylenequinone (**11**), and one benzocycloalkanone (**12**), were isolated from an ethyl acetate extract of *A. alternata*. Their chemical structures were analyzed using 1D and 2D nuclear magnetic resonance (NMR), high-resolution (HR) MS with electrospray ionization (ESI), and computational ECD. The metabolites were identified as alternatiol (**1**), phialophoriol (**2**) [34], alternariol (**3**) [35], alternariol-5-*O*-methyl ether (**4**) [36], altertenuol (**5**) [37], altenuene (**6**) [38], 2-epialtenuene (**7**) [39], (−)-altenuene (**8**) [38], 4-hydroxy-6,9-dimethylisochromen-1-one (**9**) [40], 4-hydroxy-9-(2-hydroxypropyl)-6-methylisochromen-1-one (**10**) [40], altertoxin I (**11**) [41], and 5-hydroxyscytalone (**12**) [42] (Figure 1). Alternatiol (**1**) is a new compound, and the isolation of **5**, **8**–**10**, and **12** from *A. alternata* is reported herein for the first time.

Compound **1** was isolated as a yellow amorphous powder. Based on the ^1^H and ^13^C-NMR spectral data (Table 1), its molecular formula was C_16_H_14_O_8_. The calculated *m*/*z* for the sodium adduct of **1**, [M + Na]^+^, was 357.0581. It was detected at *m*/*z* 357.0591 with ten degrees of unsaturation. ^1^H-NMR analysis of **1** revealed two hydroxy groups at δH 7.25 (s, 11-OH) and 11.12 (s, 3-OH), two meta-coupled aromatic protons at δH 7.23 (d, J = 2.0 Hz, H-6) and 6.81 (d, J = 2.0 Hz, H-4), one olefinic proton at δH 7.17 (s, H-9), two methoxy groups at δH 3.92 (s, 5-OCH_3_) and 3.61 (s, 14-OCH_3_), and one methyl group at δH 1.51 (s, H-13). In the ^13^C-NMR and heteronuclear single quantum correlation (HSQC) analysis of **1**, three carbonyl groups were detected at δC 196.5 (C-10), 168.0 (C-14), and 165.5 (C-1). Aromatic carbons were detected at δC 166.0 (C-5), 163.8 (C-3), 131.5 (C-7), 106.0 (C-6), 105.0 (C-4), and 99.6 (C-2). We identified a non-protonated sp^2^ quaternary carbon at δC 163.5 (C-8), one olefinic carbon at δC 125.8 (C-9), and two oxygenated sp^3^ quaternary carbons at δC 88.8 (C-12) and 87.5 (C-11). Two methoxy groups were detected at δC 56.4 (5-OCH_3_) and 53.0 (14-OCH_3_), and one methyl group was detected at δC 24.3 (C-13). The ^1^H and ^13^C-NMR spectra of **1** were quite similar to those of **2**, although they were not identical due to substitution on the C ring. Compound **2** is an altenusin derivative with a 6/6/5 tricyclic ring skeleton. Substitution with a methyl group in the C ring of **2** occurs at C-9, and substitution with a hydroxy group occurs at C-11. Unlike **2**, the heteronuclear multiple bond correlation (HMBC) maps between H-9 and C-7, C-8, C-10, C-12 and between H-3/H-13 and C-8/C-12, and C-11 of **1** revealed that the methyl group was attached to C-12 of the cyclopentenone ring. The presence of an α,β-unsaturated carbonyl group on the cyclopentenone moiety was evident in the HMBC correlations. The positions of the two methoxy groups were determined from the HMBC correlations between 5-OCH_3_ and C-5 and between 14-OCH_3_ and C-14. Furthermore, HMBC correlations between the hydroxyl proton and C-10, C-12, and C-14 revealed that C-11 was hydroxylated (Figure 2A). By comparison of the ^1^H and ^13^C-NMR chemical shifts of C-13 of compounds **1** and **6**–**8**, we considered the configuration is *R*. The strong ROESY (Rotating-frame overhauser spectroscopy) correlation between 11-OH and methyl group (H_3_-13) suggested that it was positioned on the same face of C ring (Figure 2B). Therefore, the absolute configuration of **1** is 11*S*, 12*R* configuration. On the basis of this result, **1** was elucidated by comparison with the experimental and calculated ECD spectra. Compound **1** showed striking similarity with experimental data (Appendix A). Based on these data, the whole structure of **1** was determined, named alternatiol.

### 2.2. Bioassays

The pyruvate dehydrogenase E1α subunit is a characteristic marker of PDH kinase activity. The phosphorylation of PDH E1α serine (Ser300) residues in AD-293 cells cultured with either **3** or **11** was quantified via immunofluorescence measurement. Cell viability was evaluated using a colorimetric MTT assay. The results showed that these tested compounds have no cytotoxicity at the concentrations tested. IC_50_ concentrations were determined by normalization against 5-mM dichloroacetic acid (DCA), which is a PDH kinase inhibitor. The amount of phosphorylated p-PDH E1α (Ser300) was reduced after treatment with **3** (Figure 3). Compound **3** precipitated during cell treatment, but no cytotoxicity was indicated. Phosphorylation of PDH E1α was reduced by **11** as well, and no morphological side effects were observed (Figure 3). 

The phosphorylation of PDHE 1α was measured by immunofluorescence and IC_50_ was calculated by normalizing against a pharmacological inhibitor control (DCA 5 mM). Compounds **3** and **11** of 100 μg/μL inhibited the PDH phosphorylation with IC_50_ values of 32.58, and 27.82 μM, respectively (Appendix A). Under the same conditions, DCA inhibited PDH phosphorylation at an IC_50_ concentration of 1 mM (Appendix A). Based on these results, compounds **3** and **11** increased PDH activity. Alternariol (**3**) is a protein kinase and xanthine oxidase inhibitor. Compound **3** exhibits cytotoxicity in L5178Y mouse lymphoma cells, [43,44] and **2** and **3** can kill human colon carcinoma cells [45]. Compound **3** and alternariol 5-*O*-methyl ether (**4**) induce cytochrome P450 1A1 activity in murine hepatoma cells and cause apoptosis [46]. The compounds **3**, **4**, altenuene (**6**), and 2-epialtenuene (**7**) also exhibit cytotoxic activity [43]. Compounds **3**, **4**, **6**, **7**, and altertoxin I (**11**) are known to be toxic to brine shrimp [47,48,49]. 

## 3. Conclusions

In this study, twelve secondary metabolites including a new compound (alternatiol; **1**), were isolated from the fungal strain *A. alternata* JS0515. To our knowledge, this is the first chemical investigation of *A. alternata* JS0515. In addition, compounds **3** and **11** increased PDH activity in AD-293 human embryonic kidney cells and significantly inhibited PDH phosphorylation. Overall, the results suggest that altenusin and isochromanone derivatives from JS0515 can possibly be used to treat some various metabolic disorders. In a future study, we will go into compound **3** and **11** induces the apoptosis of cancer cells by through inhibition of PDH phosphorylation.

## 4. Materials and Methods 

### 4.1. General Experimental Procedures

Optical rotations were determined using a Jasco DIP-370 automatic polarimeter. The HR-ESI-MS spectra were acquired on a UHR ESI Q-TOF (quadrupole time-of-flight) mass spectrometer (Bruker, Billerica, MA) and a Q-TOF micromass spectrometer (Waters, Milford, MA, USA). NMR spectra were taken in DMSO-*d*_6_, CD_3_OD, pyridine-*d*_5_, and CDCl_3_ and chemical shifts were referenced relative to the corresponding signals (δ_H_ 2.50/δ_C_ 39.50 for DMSO-*d*_6_ (compound **1**); δ_H_ 3.30/δ_C_ 49.00 for CD_3_OD (compound **2** and **6**–**10**); δ_H_ 8.73/δ_C_ 150.22 for pyridine-*d*_5_ (compound **3**–**5**); δ_H_ 7.25/δ_C_ 77.00 for CDCl_3_ (compound **11** and **12**)) (Cambridge Isotope Laboratories, Inc., Tewksbury, MA, USA) and measured on a Varian VNS 500 spectrometer (Varian, palo alto, CA, USA) and Bruker DPX 300 spectrometer (Bruker, Daltonics GmbH, Bremen, Germany). Semi-preparative HPLC were performed on a 600 controller (Waters, Milford, MA, USA) using a reversed-phase C_18_ column (Agilent Technologies ZORBAX SB-C18, Santa Clara, CA, USA, 250 × 21.2 mm; Phenomenex Luna C_18_, Torrance, CA, USA, 250 × 10 mm). Open column chromatography was accomplished over a silica gel 60 (70–230 mesh, Merck, Germany). Thin-layer chromatography (TLC) was performed on pre-coated silica gel 60 F_254_ and RP-18 F_254S_ plates (Merck, Darmstadt, Germany) using a UV detector and 10% H_2_SO_4_ reagent to visualize the compounds. All solvents used for the whole experiments were of analytical quality.

### 4.2. Isolation of The Fungal Strain

The fungal strain (JS515) was isolated from the beach vitex (*V. rotundifolia*), which was collected from a swamp in Suncheon, South Korea (34°83’79’’ N, 127°44’95’’ E) in September, 2011. Rhizome tissues were cut into small pieces (0.5 × 0.5 cm) and sterilized with 2% sodium hypochlorite for 1 min and 70% ethanol for 1 min, and then washed with sterilized distilled water. Fungal strains were cultured from plant tissues after about seven days of incubation on malt extract agar (MEA, Difco) supplemented with 50 ppm kanamycin, 50 ppm chloramphenicol, and 50 ppm Rose Bengal at 22 °C. The growing colony cut edges off and then pieces were transferred to fresh potato dextrose agar (PDA, Difco) for pure culture before being stored as 20% glycerol stocks in a liquid nitrogen tank at the Wildlife Genetic Resources Bank at the National Institute of Biological Resources (Incheon, Korea) before use.

### 4.3. Cultivation and Extraction of The Fungal Strain 

The JS0515 strain was cultivated according to two methods. The first method was cultivated on solid rice medium (80 g rice per 120 mL distilled water in a 500 mL Erlenmeyer flask was autoclaved) at room temperature. After three weeks, the fungal cultures were extracted with ethyl acetate (200 mL per Erlenmeyer flasks) in an ultrasonic sonomatic cleaning bath for 1 hour three times. The EtOAc extracts were then evaporated in vacuo. The EtOAc extracts were partitioned with *n*-hexane and acetonitrile to eliminate oily constituents. Then, a portion of aceonitrile was evaporated in vacuo to give an extract (335.5 mg). The second method was cultivated on PDB medium (12 g potato dextrose per 500 mL distilled water in a 1 L Erlenmeyer flask) at room temperature. After 3 weeks, the culture was extracted with EtOAc three times and then evaporated under reduced pressure to obtain the extract (860.0 mg).

### 4.4. Isolation of Secondary Metabolites 

The EtOAc extracts (330.0 mg) cultivated in rice medium were chromatographed with silica gel column chromatographic method using hexane-acetone gradient (*v*/*v*, 9:1 → 0:1) to yield five fractions (Fraction A–E). Fraction A was purified by HPLC with a C18 column using a gradient solvent system of H_2_O-acetonitrile (70:30 → 30:70) to obtain **2** (1.3 mg), **4** (1.3 mg), and **9** (1.2 mg). Fraction B was subjected to HPLC with a C18 column using H_2_O-acetonitrile gradient (65:35 → 30:70) to give **3** (10.8 mg) and **7** (1.5 mg). Fraction C was separated by HPLC with a C18 column using H_2_O-acetonitrile gradient (70:30 → 30:70) to yield **6** (5.0 mg), **8** (3.0 mg), and **10** (1.5 mg). Fraction D was purified by HPLC with a C18 column using a gradient solvent system of H_2_O-acetonitrile (75:25 → 30:70) to obtain **1** (2.0 mg) and **5** (3.2 mg).

The EtOAc extracts (850.0 mg) cultivated in PDB medium were subjected to a column chromatography over silica gel with elution of hexane-EtOAc-MeOH gradient (*v*/*v*/*v*, 20:1:0 → 1:1:0 → 1:1:0.5) solvent to yield seven fractions (Fraction 1–7). Fraction 4 was purified by HPLC with a C18 column using a gradient solvent system of H_2_O-acetonitrile (*v*/*v*, 50:50 → 0:100) to obtain **4** (3.6 mg). Fraction 5 was separated by silica gel column chromatography using a gradient system of hexane-acetone (*v*/*v*, 15:1 → 2:1) to give seven fractions (Fractions 5.1–5.7). Fraction 5.5 was purified by HPLC with a C_18_ column using an isocratic H_2_O-acetonitrile (*v*/*v*, 45:55) solvent system to give **3** (3.6 mg) and **11** (7.4 mg). Fraction 5.6 was isolated by HPLC with a C18 column using a gradient solvent system of H_2_O-acetonitrile (*v*/*v*, 60:40 → 0:100) to produce **6** (1.0 mg) and **12** (1.0 mg).

Alternatiol (**1**): yellow amorphous powders; UV (MeOH): λ_max_ 248, 313, 348 nm; ${\left[\alpha \right]}_{D}^{20}=-25.86$ (c 0.1, MeOH); (+) HR-ESI-MS *m/z*, 357.0591 [M + Na]^+^, calcd for C_16_H_14_O_8_Na, 357.0581; ^1^H-NMR (500 MHz, DMSO-*d*_6_) δ 11.12 (1H, s, 3-OH), 7.25 (1H, s, 11-OH), 7.23 (1H, d, *J* = 2.0 Hz, H-6), 7.17 (1H, s, H-9), 6.81 (1H, d, *J* = 2.0 Hz, H-4), 3.92 (3H, s, 5-OCH_3_), 3.61 (3H, s, 14-OCH_3_), 1.51 (3H, s, H-13); ^13^C-NMR (125 MHz, DMSO-*d*_6_) δ 196.5 (C-10), 168.0 (C-14), 166.0 (C-5), 165.5 (C-1), 163.8 (C-3), 163.5 (C-8), 131.5 (C-7), 125.8 C-9), 106.0 (C-6), 105.0 (C-4), 99.6 (C-2), 88.8 (C-12), 87.5 (C-11), 56.4 (5-OCH_3_), 53.0 (14-OCH_3_), 24.3 (C-13); ^1^H- and ^13^C-NMR (500 and 125 MHz, DMSO-*d*_6_) spectroscopic analysis (Table 1). HMBC correlations (DMSO-*d*_6_, H-#→C-#) H-4→C-2, C-3, C-5, and C-6; H-6→C-2, C-4, C-5, and C-8; H-9→C-7, C-8, C-10, C-11, and C-12; H-13→C-8, C-11, and C-12; 11-OH→C-10, C-11, C-12, and C-14; 5-OCH_3_→C-5; 14-OCH_3_→C-14.

### 4.5. Pyruvate Dehydrogenase Complex (PDH) Cellular Activity

We added 0.2 % Gelatin to the black 96-well plate with clear bottom and incubated for 1hr. After then, the plate was washed with growth media. Human AD-293 cells, derivative of the HEK293 cells, were seeded into black 96-well plates with clear bottom and grown for 24 hours. Compounds were then added and incubated for 24 hours. The cells were then fixed with 2% paraformaldehyde, permeabilized. Anti-PDHE1 pSer300 (Merk Millipore, AP1064, Darmstadt, Germany) was added and incubated overnight. Next, the cells were washed and Alexa fluor 488, goat anti-rabbit ab (Invitrogen, A11008, Waltham, MA, USA) was added with Hoechst 33258 (Invitrogen, H3569, Waltham, MA, USA) and incubated for two hours. Finally, cells were washed and the plates were measured in Operetta (PerkinElmer, Waltham, MA, USA). The raw data were normalized for the pharmacological inhibitory control (5 mM dichloroacetate (DCA)) and percent effect values using the software package Harmony High-Content Imaging and Analysis Software 3.1. The Deose response curves were generated by plotting the percent effect values and calculated IC_50_ via GraphPad Prism 6.

### 4.6. Statistical Analysis 

All values are expressed as means ± standard error of the mean. The statistical significance threshold (*p* < 0.05 for all analyses) was assessed by one-way ANOVA followed by Tukey’s post-hoc test for multiple comparisons using Prism 5.01 software (GraphPad Software Inc., San Diego, CA, USA).

## Figures and Tables

**Figure 1 molecules-24-04450-f001:**
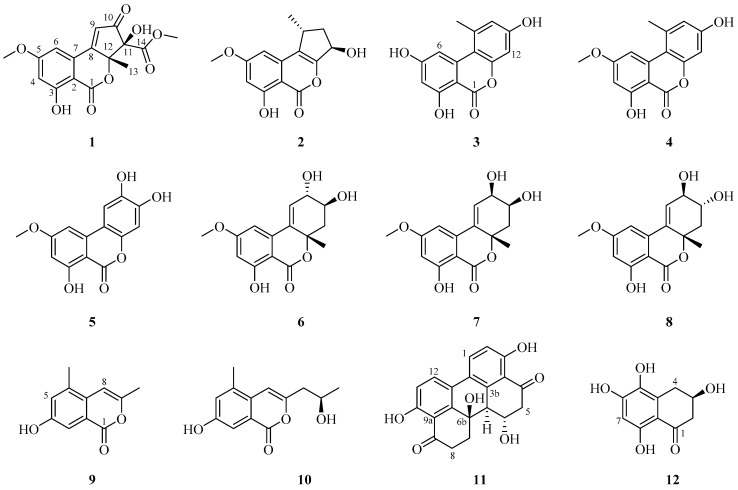
Structures of compounds **1**–**12** isolated from *A. alternata*.

**Figure 2 molecules-24-04450-f002:**
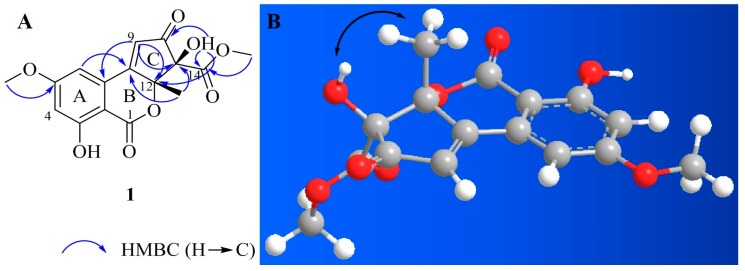
Key HMBC (**A**) and ROE (**B**) correlations of compound **1.**

**Figure 3 molecules-24-04450-f003:**
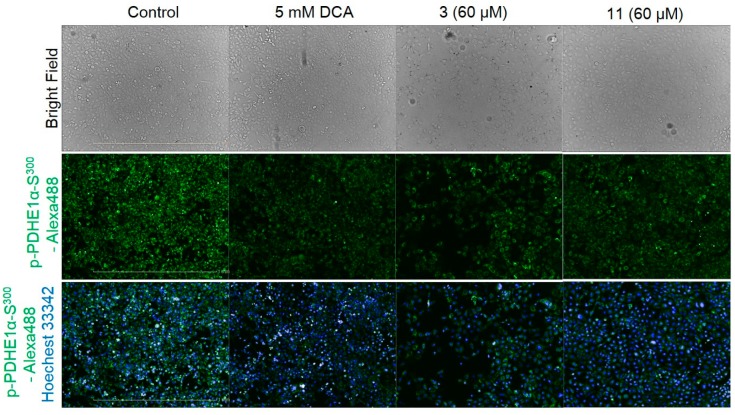
Immunofluorescence analysis of p-PDH E1α (Ser300) in AD-293 cells (green) treated with dichloroacetic acid (DCA) and compounds isolated from *A. alternata*. The cell nuclei (blue) were stained with Hoechst 33342.

**Table 1 molecules-24-04450-t001:** ^1^H (500 MHz) and ^13^C-NMR (125 MHz) spectroscopic analysis of compound **1** in DMSO-*d*_6_.

Position	δ_C_	δ_H_ (*J* in Hz)	Position	δ_C_	δ_H_ (*J* in Hz)
1	165.5		9	125.8	7.17, s
2	99.6		10	196.5	
3	163.8		11	87.5	
4	105.0	6.81, d (2.0)	11-OH		7.25, s
5	166.0		12	88.8	
5-OCH_3_	56.4	3.92, s	13	24.3	1.51, s
6	106.0	7.23, d (2.0)	14	168.0	
7	131.5		14-OCH_3_	53.0	3.61, s
8	163.5		3-OH		11.12, s

*J* values are shown in parentheses and reported in Hz. The assignments were based on ^1^H–^1^H COSY, HSQC, and HMBC experiments.

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
