# Peer review of "Secondary Metabolites of The Endophytic Fungus Alternaria alternata JS0515 Isolated from Vitex rotundifolia and Their Effects on Pyruvate Dehydrogenase Activity"

_molecules, 2019, doi:10.3390/molecules24244450_

Round 1
Reviewer 1 Report
This paper is about isolation of twelve secondary metabolites from Vitex rotundifolia Linne fil. and their effects on pyruvate dehydrogenase activity. The compound 1 was a new compound and its planar structure was elucidated by MS and NMR. But there was no discussion about the relative and absolute configurations of this new compound. So I prefer to reject this paper.
Author Response
Answer: Thank you for your important reminder. According to other reviewer's comment, we have been added the discussion of the relative configurations of new compound in the revised manuscript.
Reviewer 2 Report
Authors isolated from the fungus Alternaria alternata JS0515 twelve secondary metabolites. Compounds 3 and 11 increased pyruvate dehydrogenase (PDH) activity in AD-293 human embryonic kidney cells and significantly inhibited PDH phosphorylation. The IC50 values of both substances were 32.58 and 27.82 μM, respectively.
Unfortunately, I do not see AIM of this study. Authors not described why use AD-293 cells, and why they studied PDH? Suppression of PDH activity is associated with some diseases, but what impact has on AD-293 cells in terms of disease development, and what disease?
In Table 1 should be add in the first row "Number of compound".
Figure 2 is too small and unclear.
It is a pity that the authors did not study changes in viability of AD-293 cells under influence of compounds 3 and 11.
In References is lack of article titles.
In the article is lack of Conclusions.
Author Response
Authors isolated from the fungus Alternaria alternata JS0515 twelve secondary metabolites. Compounds 3 and 11 increased pyruvate dehydrogenase (PDH) activity in AD-293 human embryonic kidney cells and significantly inhibited PDH phosphorylation. The IC50 values of both substances were 32.58 and 27.82 μM, respectively.
Unfortunately, I do not see AIM of this study. Authors not described why use AD-293 cells, and why they studied PDH? Suppression of PDH activity is associated with some diseases, but what impact has on AD-293 cells in terms of disease development, and what disease?
Answer: We are thankful to reviewer for reminding us this crucial information. Human AD-293 cell line is a derivative of the commonly used HEK293 cell line, with improved cell adherence and plaque formation properties. HEK293 cells are human embryonic kidney cells transformed by sheared adenovirus type 5 DNA. AD-293 cell line is often used in inhibition of pyruvate dehydrogenase in vitro experiment (Inhibition of pyruvate dehydrogenase kinase 2 protects against hepatic steatosis through modulation of tricarboxylic acid cycle anaplerosis and ketogenesis. Diabetes 2016, 65, 2876-2887). This study is focus on bioactivity evaluation of single compounds from natural source. In previous study, we found that some isochroman derivatives increased pyruvate dehydrogenase activity. The structure of isolated compounds from JS0515 was similar to isochromans, so we chooing pyruvate dehydrogenase activity to test. Suppression of PDH activity is associated with various metabolic disorders, including obesity, non-alcoholic fatty liver disease, diabetes, and cancer. In future study, we will go into compound 3 and 11 induces the apoptosis of cancer cells by through inhibition of PDH phosphorylation.
1) In Table 1 should be add in the first row "Number of compound".
Answer: Thank you for your important reminder. This comment was carefully checked and corrected in the revised manuscript.
2) Figure 2 is too small and unclear.
Answer: According to reviewer's comment, we have been changed the Figure 2 in the revised manuscript.
3) It is a pity that the authors did not study changes in viability of AD-293 cells under influence of compounds 3 and 11.
Answer: This comment was carefully checked and corrected in the revised manuscript.
4) In References is lack of article titles.
Answer: This comment was carefully checked and corrected in the revised manuscript.
5) In the article is lack of Conclusions.
Answer: According to reviewer's comment, we have been added conclusion in the revised manuscript.
Reviewer 3 Report
This manuscript describes isolation of twelve secondary metabolites from A. alternata extracts and studied their pyruvate dehydrogenase (PDH) activity. However, among these metabolites, only one is a new compound and without shows PDH biological activity. More detailed biological activity test may be considered. In addition, the stereosturcture of 1 (C12 and C13) has not yet been determined. This manuscript should be revised and resubmitted. The following suggestions should be considered before resubmission.
(1) Please provide [α]D and melting point for compound 1.
(2) C3-OH signal is missing in the 1H NMR, this proton signal usually appears >10.0 ppm, please change the range of the spectra.
(3) ROSEY (Figure S5) should be discussed in the main text.
Author Response
This manuscript describes isolation of twelve secondary metabolites from A. alternata extracts and studied their pyruvate dehydrogenase (PDH) activity. However, among these metabolites, only one is a new compound and without shows PDH biological activity. More detailed biological activity test may be considered. In addition, the stereosturcture of 1 (C12 and C13) has not yet been determined. This manuscript should be revised and resubmitted. The following suggestions should be considered before resubmission.
(1) Please provide [α]D and melting point for compound 1.
Answer: Thank you for your important reminder. According to other reviewer's comment, we added the[α]D data in in the revised manuscript. However, because of the bioactivity test, we have no more amount of new compound to check the melting point.
(2) C3-OH signal is missing in the 1H NMR, this proton signal usually appears >10.0 ppm, please change the range of the spectra.
Answer: According to other reviewer’s comment, we changed the 1H NMR spectrum in the revised manuscript.
(3) ROSEY (Figure S5) should be discussed in the main text.
Answer: According to other reviewer’s comment, ROESY data have been discussed in the revised manuscript.
Reviewer 4 Report
This manuscript described the isolation and structure elucidation of 12 compounds from the endophytic fungus Alternaria alternata. One of the isolated compounds is new. The manuscript is accepted to me after the authors successfully respond to the following:
The major issue is the absolute stereochemistry of the new compound (1), the authors have to assign the configuration of C-11 and C-12 since all the stereochemistry of previously isolated compounds (2-12) was assigned. Also in the abstract, it was mentioned that the ECD is used but it is not mentioned in the discussion. The author can use ECD to determine the stereochemistry of 1.
Minor issues:
1- remove the author name (Linne fil) from the title
2-line 25 : remove from A. alternata
3- Line 57 spell out Alternaria for the first time then abbreviate it as A.
4-what is the rationale of choosing this assay, have the authors tested the extract or the fractions before the pure compounds
5- number all the carbons in compound 1 especially the stereocenters
6- remove figure 3 it does not add any new, the text is enough
7-line 147 specify which NMR solvent used for which compound
Author Response
This manuscript described the isolation and structure elucidation of 12 compounds from the endophytic fungus Alternaria alternata. One of the isolated compounds is new. The manuscript is accepted to me after the authors successfully respond to the following:
The major issue is the absolute stereochemistry of the new compound (1), the authors have to assign the configuration of C-11 and C-12 since all the stereochemistry of previously isolated compounds (2-12) was assigned. Also in the abstract, it was mentioned that the ECD is used but it is not mentioned in the discussion. The author can use ECD to determine the stereochemistry of 1.
Answer: According to reviewer's comment, we have been added in the revised manuscript.
Minor issues:
1- remove the author name (Linne fil) from the title
Answer: According to reviewer's comment, we have been changed title.
2-line 25 : remove from A. alternata
Answer: This comment was carefully checked and corrected in the revised manuscript.
3- Line 57 spell out Alternaria for the first time then abbreviate it as A.
Answer: This comment was carefully checked and corrected in the revised manuscript.
4-what is the rationale of choosing this assay, have the authors tested the extract or the fractions before the pure compounds
Answer: This study is focus on bioactivity evaluation of single compounds. In previous study, we found that some isochroman derivatives increased pyruvate dehydrogenase activity. The structure of isolated compounds was similar to isochromans, so we chooing pyruvate dehydrogenase activity to test.
5- number all the carbons in compound 1 especially the stereocenters
Answer: According to reviewer's comment, we have been added the number of compound 1.
6- remove figure 3 it does not add any new, the text is enough
Answer: This comment was carefully checked and corrected in the revised manuscript.
7-line 147 specify which NMR solvent used for which compound
Answer: This comment was carefully checked and corrected in the revised manuscript.
Round 2
Reviewer 1 Report
After revision, I prefer to accept this paper.
Reviewer 2 Report
Article was substantially corrected by the Authors.
Reviewer 3 Report
This manuscript describes isolation of twelve secondary metabolites from A. alternata extracts and studied their pyruvate dehydrogenase (PDH) activity. A number of suggestions and comments from former reviewers have been considered in this new manuscript. This manuscript is recommended for publication after minor revisions.
Please add C3-OH (proton) signal 11.12 ppm to Table 1 (page 4) and spectral data for alternatiol (1) (page 6, line 214), and also describe it in the main text (page 3, 2nd paragraph).
Reviewer 4 Report
The authors successfully responded to all my comments. I accepted the manuscript in its current status